# Safety of the Breast Cancer Adjuvant Radiotherapy in Ataxia–Telangiectasia Mutated Variant Carriers

**DOI:** 10.3390/cancers16071417

**Published:** 2024-04-05

**Authors:** Rayan Bensenane, Arnaud Beddok, Fabienne Lesueur, Alain Fourquet, Mathilde Warcoin, Marine Le Mentec, Eve Cavaciuti, Dorothée Le Gal, Séverine Eon-Marchais, Nadine Andrieu, Dominique Stoppa-Lyonnet, Youlia Kirova

**Affiliations:** 1Department of Radiation Oncology, Institut Curie, 75248 Paris, France; rayan.bensenane@aphp.fr (R.B.); alain.fourquet@curie.fr (A.F.); 2Department of Radiation Oncology, Institut Godinot, 51454 Reims, France; arnaud.beddok@reims.unicancer.fr; 3CRESTIC EA 3804, University Reims Champagne-Ardenne, 51454 Reims, France; 4Inserm U900, Institut Curie, PSL Research University, Mines ParisTech, 75248 Paris, France; fabienne.lesueur@curie.fr (F.L.); eve.cavaciuti@curie.fr (E.C.); dorothee.legal@curie.fr (D.L.G.); severine.eon-marchais@curie.fr (S.E.-M.); nadine.andrieu@curie.fr (N.A.); 5Department of Genetics, Institut Curie, 75248 Paris, France; mathilde.warcoin@curie.fr (M.W.); marinelementec@curie.fr (M.L.M.); dominique.stoppalyonnet@curie.fr (D.S.-L.); 6Inserm U830, Institut Curie, Paris-Cité University, 75248 Paris, France; 7Paris Sciences & Lettres Research University, 75248 Paris, France; 8University Versailles, 02100 St. Quentin, France

**Keywords:** breast neoplasm, radiation therapy, *ATM*, radio-induced toxicities, pathogenic variant, SNP, rs1801516

## Abstract

**Simple Summary:**

Of the worldwide population, 0.5 to 1% of people are carrying a heterozygous mutation of Ataxia–Telangiectasia Mutated (*ATM*) gene. While the clinical radiosensitivity of carriers of germline biallelic inactivation of the *ATM* gene is well described, controversies are observed for monoallelic carriers of *ATM* mutation. The aim of this study is to evaluate acute and late toxicities after adjuvant breast radiation therapy in *ATM* pathogenic variant carriers. This observational retrospective study showed an absence of significative acute and late toxicities after breast radiation therapy among patients carrying a heterozygous rare variant of the *ATM* gene. Single nucleotide polymorphism rs1801516 (G/A), described as associated with late subcutaneous fibrosis, was not associated with this late adverse event in our study.

**Abstract:**

The Ataxia–Telangiectasia Mutated *(ATM)* gene is implicated in DNA double-strand break repair. Controversies in clinical radiosensitivity remain known for monoallelic carriers of the *ATM* pathogenic variant (PV). An evaluation of the single-nucleotide polymorphism (SNP) rs1801516 (G-A) showed different results regarding late subcutaneous fibrosis after breast radiation therapy (RT). The main objective of this study was to evaluate acute and late toxicities in carriers of a rare *ATM* PV or predicted PV and in carriers of minor allele A of rs1801516 facing breast RT. Fifty women with localized breast cancer treated with adjuvant RT between 2000 and 2014 at Institut Curie were selected. Acute and late toxicities in carriers of a rare PV or predicted PV (n= 9), in noncarriers (n = 41) and in carriers of SNP rs1801516 (G-A) (n = 8), were examined. The median age at diagnosis was 53 years old and 82% of patients had an invasive ductal carcinoma and 84% were at clinical stage I–IIB. With a median follow-up of 13 years, no significant difference between carriers and noncarriers was found for acute toxicities (*p* > 0.05). The same results were observed for late toxicities without an effect from the rs1801516 genotype on toxicities. No significant difference in acute or late toxicities was observed between rare *ATM* variant carriers and noncarriers after breast RT for localized breast cancer.

## 1. Introduction

Breast cancer remains a global health challenge, accounting for over two million new cancer cases annually. Breast cancer mortality rates ranked first among all cancers in 2020, with seven hundred thousand deaths [1]. Non-metastatic breast cancer represents 70 to 80% of all breast cancers diagnosed every year, and its management is based on a multimodal approach [2]. Radiation therapy (RT) is a key part of this multimodal treatment, especially in adjuvant loco-regional breast cancer by reducing the risk of local recurrence by 50% [3]. With advances in genomic exploration over the past two decades, some germline variants have been described as associated with an increased risk of breast cancer, such as pathogenic variants (PVs) in the *BRCA1*, *BRCA2, PALB2*, *TP53* and *ATM* genes [4]. Identifying such germline PVs has major implications in daily oncologist practice, thanks to the development of specific targeted treatments such as PARP inhibitors.

The *ATM* (Ataxia–Telangiectasia Mutated) gene is located on 11q22.3. It encodes a serine/threonine kinase which acts a key regulator of signaling following DNA double-strand breaks [5]. Individuals with the biallelic *ATM* PV (either homozygous PV carriers or compound heterozygous PV carriers) present with Ataxia–Telangiectasia (A-T) syndrome. A-T patients have a hypersensitivity to ionizing radiation and agents that cause DNA double-strand breaks [6,7,8]. It is estimated that 0.5 to 1% of the general population carry an *ATM* variant classified as pathogenic for A-T disease, and studies conducted in hereditary breast and ovarian cancer families or early-onset breast cancer cases showed that such variants confer a 2 to 4-fold increase in breast cancer risk as compared to noncarriers [9,10,11]. Therefore, it is estimated that over 4% of all breast cancer cases may carry a so-called PV, which represents over 80,000 new cases per year worldwide.

Over the last decade, a number of studies have been conducted in cancer patients to assess the role of *ATM* variants as a risk factor for normal tissue complications after RT. Before clinical data, in vivo studies have shown a lower survival rate after 2 Gray (Gy) irradiation of lymphoblastic cells of monoallelic carriers of *ATM* than non-mutated donor cells [12,13]. Those data have been correlated with in vivo studies showing an increase in chromatid abnormalities after a 1 Gy irradiation of *ATM* monoallelic knock-out mice [14,15]. In addition to rare PVs, the common single-nucleotide polymorphism (SNP) rs1801516 (c.55557G>A; p.Asp1853Asn) has been investigated over the years for its potential association with late radiation-induced complications, which has led to controversial results. Three systematic reviews and meta-analyses have addressed its impact on normal tissue injuries after RT, of which two showed a significantly increased risk of acute toxicity and radiation-induced fibrosis, respectively, among carriers of the minor allele A, while another study found no association [16,17,18]. Regarding the role of rare *ATM* PVs in such toxicities, inconsistent results have been reported as well [19,20,21,22].

The primary objective of this study was to evaluate whether women with a monoallelic rare *ATM* PV or predicted PV and who underwent RT for non-metastatic breast cancer are at higher risk of acute and late toxicities than noncarriers. We also examined the effect of the rs1801516 genotype on radio-induced toxicity.

## 2. Materials and Methods

### 2.1. Study Population

The study population consisted of female breast cancer patients enrolled in two French national studies: the familial case–control study GENESIS [23] and the ongoing familial prospective cohort, CoF-AT2 [24]. GENESIS (GENE SISters) was designed to investigate familial predisposition to breast cancer. Index cases were enrolled through the national network of family cancer clinics (Genetics and Cancer Group of UNICANCER). Eligible index cases were cases diagnosed with infiltrating mammary or ductal adenocarcinoma, tested negative for *BRCA1* and *BRCA2* PVs, and had a sister with breast cancer. Female cancer-free friends or colleagues of the index cases were also enrolled and served as controls. Inclusions started in February 2007 and ended in December 2013. Clinical, epidemiological, familial data and biological samples are centralized at Institut Curie. Blood samples were collected at inclusion.

The prospective cohort CoF-AT2 was initiated in 2003 to include Ataxia–Telangiectasia (A-T) patients’ relatives. The study protocol was amended in 2017 in order to enroll new participants from cancer-prone families segregating an *ATM* pathogenic or predicted pathogenic variant (see definitions of PV and predicted PV in the next paragraph) through the national network of family cancer clinics. Epidemiological, familial and clinical data, together with biological samples of participants, were collected. A genetic test targeting the *ATM* variant identified in the index case was performed in the Department of Genetics of Institut Curie for all relatives enrolled in CoF-AT2. All blood samples were collected before RT treatment.

For the present retrospective study, we selected women from GENESIS and CoF-AT2 affected with breast cancer and those included were at least treated with breast RT for local and/or loco-regional breast cancer at Institut Curie Paris between 2000 and 2014. Exclusion criteria were women aged under 18 years, patients with metastatic breast cancer and patients with prior RT treatment involving patients’ breasts in treatment fields. Extraction of data from medical records was performed by two of the coauthors (AB and RB) and both were blinded of the *ATM* status of the patients.

### 2.2. ATM Variants Identification and Classification

In GENESIS, the entire coding sequence of *ATM* was sequenced in blood DNA of participants in the context of a large case–control study investigating the contribution of rare variants in DNA repair genes in breast cancer susceptibility. Detailed information on sequencing procedures and variant filtering and annotation is described in Girard et al.’s research [25]. For the present study, as in the original study, only loss-of-function variants (i.e., indels frameshift, stop gain, stop loss, start loss and canonical splice variants predicted to result in a truncated protein) and missense variants with a minor allele frequency below 0.05% in GENESIS controls were retained. We further filtered missense variants to keep only those predicted as deleterious using the in silico prediction tool CADD [26,27] and Align-GVGD classifier [28,29]. Align-GVGD categorizes missense substitutions into seven grades ordered from evolutionarily most likely (C0) to least likely (C65). Missense variants with a PHRED CADD score equal or above 10 and/or classified as C45, C55 or C65 by Align-GVGD were retained. In CoF-AT2, relatives of index cases were genotyped for the *ATM* variant identified in the index case. The same rules as in GENESIS were used to select eligible *ATM* variants.

In our analyses, all loss-of-function variants and missense variants classified as pathogenic for A-T disease were considered as “pathogenic variants (PVs)”, and other retained missense variants were considered as “predicted pathogenic variants (PPVs)”.

In total, forty-seven breast cancer cases from GENESIS and three breast cancer cases from CoF-AT treated by RT at the Institut Curie were included in this study; nine of them were heterozygous for an *ATM* PV or PPV (seven in GENESIS and two in CoF-AT2)**.** Among the nine carriers of a PV or PPV, five had the genotype rs1801516-GG, one had the genotype rs1801516-GA, and the rs1801516 genotype was not available for three of them. Among the 41 noncarriers of a PV or PPV, 25 had the genotype rs1801516-GG, 7 had the genotype rs1801516-GA and the rs1801516 genotype was not available for nine of them.

### 2.3. Radiation Therapy Treatment Characteristics and Follow-Up

All women underwent a dosimetric computed tomography (CT) scanner without injection in the radiation therapy department of Institut Curie Paris in order to prepare the RT plan of treatment. Delineation of clinical tumor volume (CTV) and organ at risk (OAR) was performed following international recommendations. Dose prescriptions were under clinician appreciation and ranged between 45 Gy and 50 Gy for whole breast irradiation and up to 14 to 16 Gy when a sequential boost was performed on the tumoral surgery bed. Each patient had weekly clinics during the RT. Then, all patients were followed up with every three months with clinical examination and annual ultrasound and mammograms. Acute toxicities were defined as appearance of dermatitis, dysphagia or lymphoedema in the three months after RT. Late toxicities were defined as subcutaneous fibrosis, telangiectasia, lymphoedema or heart disease occurring more than three months after the end of RT treatment. Toxicities were graduated using the CTCAE v.5 scale [30].

### 2.4. Statistical Analysis

Follow-up was calculated from the date of the end of RT to the date of last news. The median follow-up was estimated by the inverted Kaplan–Meier method. Baseline characteristics were summarized as numbers and percentages for qualitative data and as means and standard deviations or medians with the minimum and maximum (or inter-quartile range) for continuous variables. The Chi 2 test or Fisher test was used for the analysis of the contingency tables. The risk of late toxicity, such as fibrosis, was assessed within a competing risks framework, recognizing death as a significant competing event. Cumulative incidence functions were used for this analysis, and a Fine–Gray model was implemented to appropriately handle the complexity introduced by competing risks. All *p*-values were two-sided, and a 5% level of significance was used. Analyses were carried out using software R 4.2.2. (http://cran.r-project.org; accessed on 1 October 2022).

## 3. Results

### 3.1. Population Characteristics

The median follow-up was 13 years (range 1.6 to 21.9). In this study, 50 patients (9 *ATM* rare variant carriers and 41 noncarriers) have been analyzed and their main characteristics are summarized in Table 1. *The* rare *ATM* variant carriers and noncarrier patients had the same baseline characteristics. The median age was 53 years old (range: 35–77) and the median Body Mass Index (BMI) was 24.6 kg/m^2^ (range: 18.2–34.9). More than two-thirds of the patients had no antecedent of current and/or former smoking history (70%). Only one patient had a skin disorder which was diagnosed as psoriasis and was not clinically present in the RT area. Repartition of right or left breast cancer was equivalent. The most represented tumor localization in the breast was in the External Upper Quadrant (27/50, 54%), followed by the Union of Upper Quadrant (8/50, 16%) and the Internal Upper Quadrant (6/50, 12%). The main clinical stage (as described according to the American Joint Committee on Cancer (AJCC) 2017 v8 guidelines) was stage II (22/50, 44%) [31]. In total, 82% of the tumors were invasive ductal, overexpressing the Estrogen Receptor (ER) in 80% of cases, and without presence of embolus for two-thirds of them.

### 3.2. Characteristics of Genetic Variants

The rare *ATM* variants identified in the investigated series are described in Table 2. Among the nine rare variant carriers, three had a pathogenic variant, five a variant of unknown signification (VUS) and one had a benign variant according to the ClinVar classification.

Additionally, eight patients were heterozygous for the minor allele A of rs1801516 (c.55557G>A; p.Asp1853Asn). Information regarding this common variant was missing for four women. Of note, one patient with the genotype rs1801516-GA also carried the rare variant c.4853G>A.

### 3.3. Treatment Details

The treatment characteristics are given in Table 3. The two groups had no significative difference between the received treatments. Eighty percent (40/50) of patients underwent a breast conserving surgery and sixty-eight percent (34/50) had an axillary lymph node dissection (ALND). Data on axillary surgery were not available for four patients. None of the patients had an immediate breast reconstruction in case of mastectomy. Neo-adjuvant treatment was performed for nine patients (18%) and consisted of Epirubicine/Cyclophosphamide (EC)-Taxotere, 5FU-Navelbine or FEC100 protocols. Residual Cancer Burden score (RCB score) was evaluated as RCB0 for two patients (22%), RCB I and II for one patient (11%), respectively, and RCB III for five patients (56%).

Adjuvant treatment was delivered to 22 patients (44%); it was composed of adjuvant chemotherapy (36%) or targeted therapy (8%) with Trastuzumab. Adjuvant hormonotherapy was delivered to 32 patients (64%). One patient had an ovariectomy for the purposes of anti-hormonal treatment.

Regarding RT characteristics, 3D-CRT (Conformal RT) was used for 70% of patients and Isocentric lateral decubitus (ILD) for 30% of treated women. The mean dose of RT was 56 Gy (range: 45 to 71 Gy) and the mean RT duration was 42.7 days (range: 28 to 81). Three different RT sources were used: ^60^Cobalt (18/50, 36%), X-ray (13/50, 26%) or a mixed treatment of ^60^Cobalt and electron (2/50, 4%) or X-ray and electron (17/50, 34%). Only two patients had concomitant chemotherapy during RT. The treated volumes of RT were as follows: whole breast irradiation in 44% (22/50), or whole breast associated with RT of lymph nodes (area I–I–III–IV ± IMNI) (28/50, 56%), with or without a boost of the tumor bed (22/50, 44%).

### 3.4. Acute Toxicities

The acute toxicities of rare monoallelic *ATM* PV or predicted PV carriers are presented in Table 4. Over the nine carriers, the main acute toxicity was dermatitis for eight of them (89%). No significative difference between *ATM* PV or predicted PV carriers and noncarriers was observed when looking at acute dermatitis (*p* = 0.98), dysphagia (*p* = 1) or lymphoedema (*p* = 1) (Figure 1). The group of rare monoallelic *ATM* PV carriers was composed of three patients, of which two experienced grade 1 dermatitis and one woman experienced grade 2. No other acute toxicity was found.

We next compared the manifestation of acute toxicities between the group of eight patients with the genotype rs1801516-GA and the group of 25 patients with the genotype rs1801516-GG and not carrying a rare PV or predicted PV. After a median follow-up of 13 years, no significative difference regarding acute dermatitis (*p* = 0.23), dysphagia (*p* = 1) and lymphoedema (*p* = 0.88) was demonstrated (Appendix A). The unique patient carrying both genotype rs1801516-GA and the predicted PV c.4853G>A presented grade 1 acute dermatitis and no other acute toxicity.

When focusing on the 16 patients carrying either a rare monoallelic PV or predicted PV and/or the minor allele A of SNP rs1801516, the main acute toxicity was dermatitis, impacting 75% of women, representing twelve women above this subgroup. Regarding acute toxicities, no significant difference was observed between the carriers of a rare monoallelic PV or predicted PV *ATM* carriers and carriers of the minor allele A of SNP rs1801516 as compared to noncarriers (dermatitis: *p* = 0.65/ dysphagia: *p* = 1/ lymphoedema: *p* = 1) (Appendix A).

### 3.5. Late Toxicities

Late toxicities for the nine rare *ATM* PV or predicted PV carriers are detailed in Table 4. The main late toxicity was subcutaneous fibrosis for two out of nine patients (25%). With a median follow-up of 13 years, no significant difference was found for subcutaneous fibrosis (*p* = 0.16), telangiectasia (*p* = 0.33), lymphoedema (*p* = 0.72) and heart disease (*p* = 1) (Figure 2). No grade 3 and higher late toxicities were reported in both groups. No late plexopathy was found. The group of rare monoallelic *ATM* PV carriers was composed of three patients who did not experience any late toxicity.

With a median follow-up of 13 years, 50% (4/8) of patients with the genotype rs1801516-GA experienced late subcutaneous fibrosis, but this proportion did not differ in the group of patients with the genotype rs1801516-GG (*p* = 0.87) (Appendix A). No significant difference was observed for other late RT-related complications such as late lymphoedema (*p* = 0.65), late telangiectasia (*p* = 1) and late heart disease (*p* = 1). Late subcutaneous fibrosis did not differ between carriers of a rare *ATM* PV or predicted PV and/or minor allele A of rs1801516 and noncarriers (*p* = 0.67) (Appendix A).

## 4. Discussion

In 2017, the National Comprehensive Cancer Network (NCCN) made recommendations regarding adjuvant RT of breast cancer for women carrying a monoallelic *ATM* pathogenic variant. The NCCN recommends not to avoid adjuvant RT, which is consistent with our present results showing an absence of supplementary acute or late toxicities in this specific population [32].

Our retrospective study is one of the most recent studies with long term follow-up evaluating the incidence of acute and late radiation toxicities in women with breast cancer and heterozygous for a rare PV or predicted PV or heterozygous for allele A of the common SNP rs1801516. Our results are consistent with those of Bremer et al., showing no difference in acute and late cutaneous toxicity after treatment for early breast cancer with 3D-CRT between a group of ten patients carrying a known *ATM* PV leading to a frameshift or missense [20].

In the present work, we focused our analyses not only on carriers of a rare variant reported as pathogenic for A-T or classified as pathogenic for HBOC in ClinVar but we also included a few patients carrying a predicted PV according to in silico tools. This is because such variants have also been associated with breast cancer in large case–control studies, and the risk estimates are close to the risk estimates associated with the so-called PV [25,29]. National and international initiatives are ongoing to clarify the role of such variants and of other VUSs identified through multi-gene panel testing [33,34,35,36].

Moreover, we did not observe an association between *ATM* variant status of the patient and clinical stage and histology criteria associated with aggressive tumors (for example, embolus, high mitotic index, triple-negative histological subtype), which is interesting in the discussion of the adjuvant RT treatment schedule of carriers of *ATM* PV or PPV who do not seem to have more aggressive histological characteristics than noncarrier patients. An exploration of a possible difference in histological subtypes of breast cancer in *ATM* variant carriers compared to other genes implicated in DNA double-strand break repair, such as *BRCA1* and *BRCA2*, was conducted by Abdulrahman et al. in 2018 and showed that tumors of *ATM* VUS carriers seem smaller, with lower pathologic T stages at diagnosis and greater surrogate molecular subtypes [37]. In a previous study, we performed a systematic pathology review of breast tumors from 21 ATM PV carriers from A-T families and 18 PV or predicted PV carriers from Hereditary Breast and Ovary Cancer families (including patients enrolled in CoF_AT2 and GENESIS), and we found that *ATM*-associated breast tumors belong mostly to the luminal B subtype in a retrospective tumor [38].

One relevant piece of information from our study is the exploration of the well-studied SNP rs1801516 (c.5557G>A, p.Asp1853Asn), leading to controversy especially regarding late subcutaneous fibrosis after RT. Andreassen et al. investigated seven patients with the heterozygous genotype of rs1801516 and showed a significant association between this SNP and grade 3 subcutaneous fibrosis after breast RT at 50 Gy dose (incidence in individuals with the GA genotype: 37% vs. incidence in individuals with the GG genotype: 16%, *p* = 0.03) [19]. Their results are in accordance with the results of the study by Ho et al. including 131 patients, with 15 of them carrying the A allele. In the latter study, 53% of the carriers and 27% of the noncarriers had grade 2 to 4 subcutaneous side effects (OR: 3.1, 95% CI 1.1–9.4) [22]. Our results do not confirm these findings, but this may be due to a lack of power given our limited sample.

This study has several limitations: the retrospective analysis which is associated with a risk of bias and especially a survivorship bias since women are the prevalent cases and the small sample of patients carrying a well-known *ATM* PV, thus rendering power of statistical analysis quite unsatisfactory for this particular population. Other limitations are the technique of RT (majority of 3D-CRT) and the type of energy used (^60^Cobalt), which is not the standard treatment nowadays for RT for breast cancer with the outcome of IMRT (Intensity Modulated Radiotherapy). Nonetheless, our findings could be interesting in daily practice, given the few adverse events observed in this population, especially in long-term follow-up of patients treated with 3D-CRT and ^60^Cobalt, which are more associated with adverse events.

## 5. Conclusions

This study showed no association between rare *ATM* PVs or predicted PVs and manifestation of acute or late toxicities after breast RT for localized breast cancer in heterozygous variant carriers. In this small series of patients, the minor allele A of rs1801516 was not associated with late subcutaneous fibrosis. Further larger and prospective investigations are needed to confirm our findings in order to better personalize RT for patients carrying a monoallelic alteration of *ATM*.

## Figures and Tables

**Figure 1 cancers-16-01417-f001:**
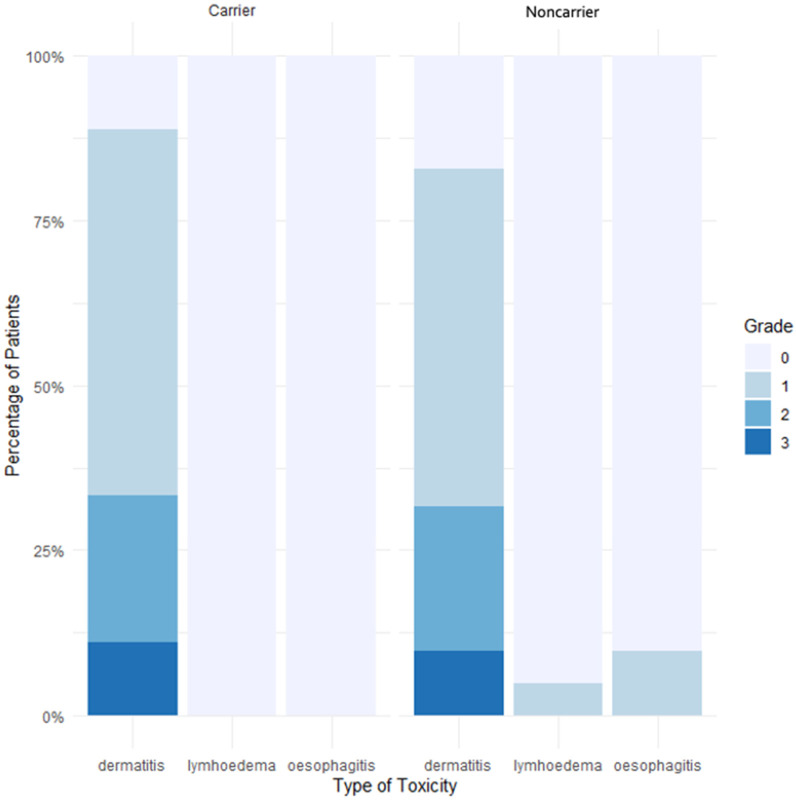
Percentage of acute toxicities in carriers and noncarriers of a rare *ATM* PV or predicted PV. Abbreviations: Carrier: patients with a rare *ATM* pathogenic variant (PV) or predicted PV, Noncarrier: patients with no rare *ATM* PV or predicted PV.

**Figure 2 cancers-16-01417-f002:**
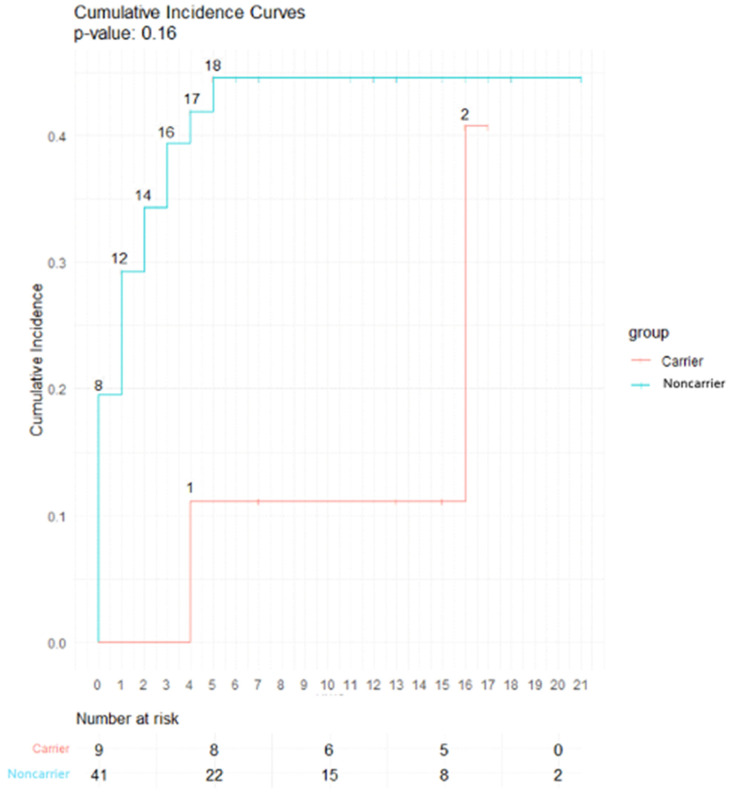
Incidence curve of late subcutaneous fibrosis in carriers and noncarriers of a rare *ATM* PV or predicted PV. Abbreviations: Carrier: patients with a rare *ATM* Pathogenic variant (PV) or predicted PV, Noncarrier: patients with no rare *ATM* PV or predicted PV.

**Table 1 cancers-16-01417-t001:** Baseline characteristics of the 50 breast cancer patients included in the analyses.

Characteristics	Total Number n = 50	Rare *ATM* Variant Carriers * n = 9	Noncarriers n = 41	*p*-Value
Median age at diagnosis (years, [median, range])	53 (35–77)	52 (36–77)	53 (35–75)	0.48
Mean follow up (years, range)	13.1 (1.6–21.9)	14.3 (7–18.2)	12.9 (1.6–21.9)	0.53
BMI (kg/m^2^, [median, range])	24.6 (18.2–34.9)	23.4 (18.2–28.9)	22.9 (18.2–34.9)	0.48
Smoking status				
Ever (30%)	15	2	13	
Never (70%)	35	7	28	0.87
Histopathology (WHO classification)				
In situ carcinoma (4%)	2	0	4	
Invasive ductal carcinoma (82%)	41	9	32	
Invasive lobular carcinoma (14%)	7	0	7	0.29
Immunochemistry				
Grade				
1 (26%)	13	1	12	
2 (44%)	22	5	17	
3 (26%)	13	3	10	
Unknown (4%)	2	0	2	0.58
Mitotic index (%)				
<4 (20%)	10	2	8	
4–12 (44%)	22	5	17	
>12 (30%)	15	2	13	
Unknown (6%)	3	0	3	0.65
Embolus				
Yes (26%)	13	4	9	
No (68%)	34	5	29	
Unknown (6%)	3	0	3	0.30
ER expression				
Positive (80%)	40	8	32	
Negative (16%)	8	1	7	
Unknown (4%)	2	0	2	0.70
HER 2 expression				
0 (40%)	20	3	17	
+ (18%)	9	2	7	
++ and FISH negative (6%)	3	1	2	
+++ (12%)	6	1	5	
Unknown (24%)	12	2	10	0.70
Clinical stage (AJCC 2017 v8)				
0 (4%)	2	0	2	
I (40%)	20	4	16	
IIA (24%)	12	1	11	
IIB (20%)	10	3	7	
IIIA (6%)	3	0	3	
IIIB (2%)	1	1	0	
Unknown (4%)	2	0	2	0.16

Abbreviations: *ATM: Ataxia–Telangiectasia mutated* gene, BMI: Body Mass Index, ER: Estrogen Receptor, FISH: Fluorescence in situ hybridization, HER2: Human epidermal growth factor receptor 2. * Only one patient carried a predicted pathogenic variant (PPV) and the genotype rs1801516-GA.

**Table 2 cancers-16-01417-t002:** Description of *ATM* variants identified in patients. All patients are heterozygous variant carriers.

Patient	Study	*ATM* Rare Variant	MAF in GnomAD ^a^	In Silico Predictions		Our Classification for This Study	ClinVar Classification	Genotype for rs1801516 (c.5557G>A; p.Asp1853Asn) ^d^
				CADD Phred Score ^b^	Align-GVGD ^c^			
Pt 1	GENESIS	c.8494C>T; p.Arg2832Cys	0.0004	32	C45	PV	PV	GG
Pt 2	CoF-AT2	c.3894dupT; p.Ala1299CysfsX3	0.000007	n/a	n/a	PV	PV	Unknown
Pt 3	CoF-AT2	c.5644C>T; p.Arg1882X	Not reported	36	n/a	PV	PV	Unknown
Pt 4	GENESIS	c.4709T>C; p.Val1570Ala	0.0007	17.66	C0	PPV	VUS	GG
Pt 5	GENESIS	c.2T>G; p.Met1Arg (START loss)	Not reported	23.4	C65	PPV	VUS	GG
Pt 6	GENESIS	c.6059G>T; p.Gly2020Val	Not reported	27.3	C65	PPV	VUS	GG
Pt 7	GENESIS	c.4853G>A; p.Arg1618Gln	0.00003	22.6	C0	PPV	VUS	GA
Pt 8	GENESIS	c.6067G>A; p.Gly2023Arg	0.0024	28.9	C25	PPV	VUS	GG
Pt 9	GENESIS	c.1073A>G; p.Asn358Ser	0.00001	14.04	C0	PPV	Benign	Unknown
Pt 10	GENESIS	-	-	-	-	-	-	GA
Pt 11	GENESIS	-	-	-	-	-	-	GA
Pt 12	GENESIS	-	-	-	-	-	-	GA
Pt 13	GENESIS	-	-	-	-	-	-	GA
Pt 14	GENESIS	-	-	-	-	-	-	GA
Pt 15	GENESIS	-	-	-	-	-	-	GA
Pt 16	GENESIS	-	-	-	-	-	-	GA

Abbreviations: PV: Pathogenic variant, PPV: Predicted pathogenic variant, VUS: Variant of unknown clinical significance, MAF: Minor allele frequency. ^a^ MAF in gnomAD non-Finnish European population. ^b^ CADD PHRED score using CADD v1.6. ^c^ Align-GVGD classifier categorizes missense substitutions into seven grades ordered from evolutionarily most likely (C0) to least likely (C65) (24, 25) ^d^ Align-GVGD class: C0; CADD PHRED score: 23.4; ClinVar classification: Benign; MAF in gnomAD non-Finnish European population: 0.1435.

**Table 3 cancers-16-01417-t003:** Treatment details.

Treatment Characteristics	Number of Treated Patients	% of Total	Rare *ATM* Variant Carriers	Noncarriers	*p*-Value
Type of surgery before RT					
Breast conserving	40	80	6	34	
Mastectomy	10	20	3	7	
Sentinel lymph node	12	24	2	10	
ALND	34	68	6	28	0.51
Chemotherapy					
NAC	9	18	3	6	0.39
Adjuvant chemotherapy	18	36	4	14	0.84
Targeted adjuvant therapy	4	8	0	4	0.76
Hormonotherapy					
AI	10	20	0	10	
Tamoxifen	22	44	5	17	0.37
Median duration of HT (years, range)	5 (1–11)	NA	5 (4–11)	5 (1–10)	
Technique of radiation therapy					
3D-CRT	35	70	5	30	
ILD	15	30	4	11	0.52
Mean dose of radiation therapy (Gy, range)	56 (45–71)	NA	57 (45–66)	56 (45–71)	0.64
Mean RT duration (days, range)	42.7 (28–81)	NA	42 (32–52)	43 (28–81)	0.73
Irradiated volume					
Only breast	22	44	3	19	
Breast and lymph nodes	28	56	6	22	
Associated boost	22	44	4	18	0.73

Abbreviations: AI: Aromatase inhibitors, ALND: Axillary lymph node dissection, 3D-CRT: 3D conformal radiation therapy, Gy: Gray, HT: Hormonotherapy, ILD: Isocentric lateral decubitus, NA: Not available, NAC: Neo-adjuvant chemotherapy, RT: Radiation therapy.

**Table 4 cancers-16-01417-t004:** Occurrence of acute and late toxicities after radiation therapy among rare ATM variant carriers and noncarriers.

	Number of Patients with Toxicities (%)					*p*-Value
Acute toxicities						
		Grade 0	Grade 1	Grade 2	Grade 3	
Dermatitis	Noncarriers	7 (14%)	21(42%)	9 (18%)	4 (8%)	
	Carriers	1 (2%)	5 (10%)	2 (4%)	1 (2%)	
	Total	8 (16%)	26 (52%)	11 (22%)	5 (10%)	0.98 (Chi 2)
Dysphagia	Noncarriers	37 (74%)	4 (8%)	0	0	
	Carriers	9 (18%)	0	0	0	
	Total	46 (92%)	4 (8%)	0	0	1 (Chi 2)
Lymphoedema	Noncarriers	39 (78%)	2 (4%)	0	0	
	Carriers	9 (18%)	0	0	0	
	Total	48 (96%)	2 (4%)	0	0	1 (Chi 2)
Late toxicities						
		Grade 0	Grade 1	Grade 2	Grade 3 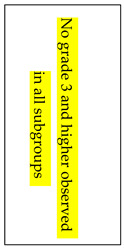	
Subcutaneous fibrosis	Noncarriers	23 (46%)	16 (32%)	2 (4%)		
	Carriers	7 (14%)	2 (4%)	0		
	Total	30 (60%)	18 (36%)	2 (4%)		0.16 (Gray)
Telangiectasia	Noncarriers	37 (74%)	4 (8%)	0		
	Carriers	9 (18%)	0	0		
	Total	46 (92%)	4 (8%)	0		0.33 (Gray)
Lymphoedema	Noncarriers	38 (76%)	0	3 (6%)		
	Carriers	8 (16%)	1 (2%)	0		
	Total	46 (92%)	1 (2%)	3 (6%)		0.72 (Gray)
Heart disease	Noncarriers	40 (80%)	1 (2%)	0		
	Carriers	9 (18%)	0	0		
	Total	49 (98%)	1 (2%)	0		1 (Gray)

## Data Availability

The data underlying this article will be shared on reasonable request to the corresponding author.

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
