# Peer review of "Safety of the Breast Cancer Adjuvant Radiotherapy in Ataxia–Telangiectasia Mutated Variant Carriers"

_cancers, 2024, doi:10.3390/cancers16071417_

Round 1

Reviewer 1 Report

Comments and Suggestions for Authors

The paper adds new information on ATM mutations variants in breast cancer radiotherapy  because few data are still available on this issue ; moreover it clarify the  effects of several defined  VUS in ATM mutations in breast cancer radiotherapy.

Author Response

Reviewer 1 comments:

The paper adds new information on ATM mutations variants in breast cancer radiotherapy because few data are still available on this issue ; moreover it clarify the  effects of several defined  VUS in ATM mutations in breast cancer radiotherapy.

=> Thank you for your comment.

Reviewer 2 Report

Comments and Suggestions for Authors

ATM accounts for germline variants in breast cancer cases, which inherit in the family. Author in the manuscript only on single ATM gene, though there are other gene also involves in hereditary breast cancer. Alter ATM gene affect the repair mechanism  in cell or cancer cell and its relation with radiotherapy needs to be proved.

Comments

1. This is kind for case presentation with ATM PV those treated with RT. Author have not described the results here and have not concluded the results. 

2. The methodology part is written long and not explaining the exact time point of sample collection in case and control. 

3. Author should explain the exclusion and inclusion criteria. Why, >18y and non metastatic cases 

4. In introduction author should have explain the relation between ATM and RT therapy. 

5. Are selected cases also has mutation in other genes ?

Author Response

Reviewer 2 comments :

  1. ATM accounts for germline variants in breast cancer cases, which inherit in the family. Author in the manuscript only on single ATM gene, though there are other gene also involves in hereditary breast cancer. Alter ATM gene affect the repair mechanism  in cell or cancer cell and its relation with radiotherapy needs to be proved.

=> Thank you for your comment. In this study, we focused our research on women with ATM monoallelic rare mutations because scarce data are available regarding this particular rare (under 1 % of worldwide population of carriers) and especially the relation between radiation therapy treatment and the impact in normal tissue; by the extend of acute and late toxicities outcome; which may have a huge impact in post-cancer life of treated patients. None of the included and analyzed patients from GENESIS study, carriers of a rare ATM PV (pathogenic variant) or PPV (predicted pathogenic variant), had a concomitant mutation in major gene implicated in hereditary breast cancer and especially BRCA1 and BRCA2 genes (line 100: “tested negative for BRCA1 and BRCA2 PV”). For women included from Co-FAT2 study, only analyses on “ATM variant identified in the index case have been performed” (line 111). This genetic analyses did not have for objective to analyze the whole potential genomic alterations eventually associated with hereditary breast cancer but only variant of ATM of Ataxia-Telangiectasia patient’s relative.

The link between mutation on ATM gene, deficient cell repair mechanism and impact on radiation therapy treatment have been well described in vitro and in vivo studies over decades. We added in the Introduction part those data : “Before clinical data, in vivo studies have shown a lower survival rate after 2 Gy irradiation of lymphoblastic cells of monoallelic carriers of ATM mutation than non-mutated donor cells [12,13]. Those data have been correlated with in vivo studies showing an increase of chromatid abnormalities after a 1 Gy irradiation of ATM monoallelic knock-out mice [14,15]. » (line 76 to 80).

  1. This is kind for case presentation with ATM PV those treated with RT. Author have not described the results here and have not concluded the results. 

   =>     Thank you for your comment. Rare ATM PV patients were represented in our study by three patients, that’s why we do not realized specific statistical analyzes above this small group of patient which could lead to a lack of power of those analysis if performed. However, thanks to your comment, we update our manuscript by adding observational data of those three patient by adding those results Acute toxicities and Late toxicities part: “The group of rare monoallelic ATM PV carriers composed of three patients have experienced a grade 1 dermatitis for two of them and a grade 2 for only one women. No other acute toxicity have been found. » (line 264 to 266) and “The group of rare monoallelic ATM PV carriers composed of three patients do not have experienced any late toxicity. » (line 300 to 302). Those results made a part of our conclusions : “This study showed no association between rare ATM PV or predicted PV and manifestation of acute or late toxicities after breast RT for localized breast cancer in heterozygous variant carriers.” (line 373 to 375).

  1. The methodology part is written long and not explaining the exact time point of sample collection in case and control. 

=>        Thanks to your comment, we added this specific information regarding exact time point of sample collection which was made at inclusion time of cases and control patients, and all sample collection were available before initiation of radiation therapy treatment. “Blood sample were collected at inclusion.” (line 103 to 104) and “All blood sample have been collected before RT treatment.” (line 112 to 113).

  1. Author should explain the exclusion and inclusion criteria. Why >18y and non metastatic cases ?

=>        Thank you for your comment. Patients under 18 years-old were not included in national studies GENESIS and CoFAT2 were patients were firstly enrolled and by stander could not have been enrolled in our retrospective study. Moreover, only localized and locoregional advanced breast cancer have been included and analyzed in our study because the main objective of our work was to evaluate loco-regional acute and late toxicities of adjuvant breast radiation therapy and between 2000 and 2014 (period of patients’ treatment), the treatment of primary breast cancer site for metastatic diseases was not well established because of scarce available data.

  1. In introduction author should have explain the relation between ATM and RT therapy. 

=>        Thanks to your comment, we develop this part in the Introduction: “Before clinical data, in vivo studies have shown a lower survival rate after 2 Gy irradiation of lymphoblastic cells of monoallelic carriers of ATM mutation than non-mutated donor cells [12,13]. Those data have been correlated with in vivo studies showing an increase of chromatid abnormalities after a 1 Gy irradiation of ATM monoallelic knock-out mice [14,15]. » (line 76 to 80).

  1. Are selected cases also has mutation in other genes ?

 => Thank you for your comment. As precised in Materials and methods part, none of the included and analyzed cases from GENESIS study, carriers of a rare ATM PV (pathogenic variant) or PPV (predicted pathogenic variant), had a concomitant mutation in major gene implicated in hereditary breast cancer and especially BRCA1 and BRCA2 genes (line 100: “tested negative for BRCA1 and BRCA2 PV”). For women included from Co-FAT2 study, only analyses on “ATM variant identified in the index case have been performed” (line 111).

Reviewer 3 Report

Comments and Suggestions for Authors

The article titled " Safety of the breast cancer adjuvant radiotherapy in ATM variant carriers" by Rayan Bensenane et al. offers valuable insights. However, there are several areas that require attention:

1. The authors should provide their own justification for the study, as previous publications have already explored the relevance of the topic. Examples of such publications include articles in PubMed, such as Int J Radiat Oncol Biol Phys. 2021 Aug 1;110(5):1373-1382. doi: 10.1016/j.ijrobp.2021.01.045; J Natl Cancer Inst. 2020 Dec 14;112(12):1275-1279. doi: 10.1093/jnci/djaa031; J Natl Cancer Inst. 2010 Apr 7;102(7):475-83. doi: 10.1093/jnci/djq055; Int J Radiat Biol. 2017 Oct;93(10):1121-1127. doi: 10.1080/09553002.2017.1344363; JCO Precis Oncol. 2021 Jan 19;5:PO.20.00334. doi: 10.1200/PO.20.00334; Radiother Oncol. 2004 Sep;72(3):319-23. doi: 10.1016/j.radonc.2004.07.010; Pract Radiat Oncol. 2024 Jan-Feb;14(1):e29-e39. doi: 10.1016/j.prro.2023.09.001; Int J Radiat Oncol Biol Phys. 2007 Nov 1;69(3):677-84. doi: 10.1016/j.ijrobp.2007.04.012, among others. Consequently, the study does not provide any innovative information.

2. The sample size of the study was comparatively small, which could limit the precision of the results

3. The authors did not consider other potential risk factors that may contribute to acute or late toxicities, including age, comorbidities, or treatment-related factors

4. There was no evaluation of more uncommon ATM PVs or SNPs that might influence radiosensitivity was done.

5. The prolonged follow-up time and retrospective analysis may add bias or confounding factors that were not taken into regard.

6. The authors did not provide data on the specific types or severity of acute and late toxicities that were evaluated.

Comments on the Quality of English Language

Moderate editing of English language required

Author Response

Reviewer 3 comments:

The article titled " Safety of the breast cancer adjuvant radiotherapy in ATM variant carriers" by Rayan Bensenane et al. offers valuable insights. However, there are several areas that require attention:

  1. The authors should provide their own justification for the study, as previous publications have already explored the relevance of the topic. Examples of such publications include articles in PubMed, such as Int J Radiat Oncol Biol Phys. 2021 Aug 1;110(5):1373-1382. doi: 10.1016/j.ijrobp.2021.01.045; J Natl Cancer Inst. 2020 Dec 14;112(12):1275-1279. doi: 10.1093/jnci/djaa031; J Natl Cancer Inst. 2010 Apr 7;102(7):475-83. doi: 10.1093/jnci/djq055; Int J Radiat Biol. 2017 Oct;93(10):1121-1127. doi: 10.1080/09553002.2017.1344363; JCO Precis Oncol. 2021 Jan 19;5:PO.20.00334. doi: 10.1200/PO.20.00334; Radiother Oncol. 2004 Sep;72(3):319-23. doi: 10.1016/j.radonc.2004.07.010; Pract Radiat Oncol. 2024 Jan-Feb;14(1):e29-e39. doi: 10.1016/j.prro.2023.09.001; Int J Radiat Oncol Biol Phys. 2007 Nov 1;69(3):677-84. doi: 10.1016/j.ijrobp.2007.04.012, among others. Consequently, the study does not provide any innovative information.

=>        Thank you for your comment. Our study seems to be innovative for two major points : first, the analyzed patient were carriers of rare ATM pathogenic (PV) or predicted pathogenic (PPV) which was not the same population describe in other studies fitting this topic while they included more common genetic SNP (such as rs1801516 such as us). Moreover, our study aim to provide a great long-term follow-up with 13 years of median follow-up which was not achieve in other studies. This particularity seems to be an innovative information regarding the potential impact in very long-term issues of breast radiation therapy in such a particular genetic mutated population represented by ATM monoallelic carriers.

  1. The sample size of the study was comparatively small, which could limit the precision of the results.

=>        Thank you for your comment. We performed precise analyses of data from single center (Institute Curie Paris) which leads in fact to a small sample size of mutated patients for analysis. However, this first retrospective study was aiming to be a first step before the realization of a French national cohort study of rare ATM monoallelic performing adjuvant breast RT, which seems to always be relevant despite no difference in term of acute and late toxicities have been found in this first study, maybe because of a lack of power especially in ATM PV carriers.

  1. The authors did not consider other potential risk factors that may contribute to acute or late toxicities, including age, comorbidities, or treatment-related factors.

=>        Thank you for your comment. In front of an absence of significative difference over acute and late toxicities between rare ATM monoallelic carriers and wild-type women, we consider that realization of multivariate analyses will not be suitable in this particular case because of the small number of case patients.

  1. There was no evaluation of more uncommon ATM PVs or SNPs that might influence radiosensitivity was done.

=>        Thank you for your comment. In the genetical analysis performed over cases of GENESIS cohort, all ATM coding sequence have been explored searching for loss-of-function and missense variants and reported monoallelic missense mutations included were the only one nowadays predicted to be in silico deleterious. In CoFAT2 genetic analyses, “only ATM variant identified in the index case have been explored” (line 135 to 136), leading eventually to a miss of other eventual SNPs or monoallelic mutation that could influence radiosensitivity.

  1. The prolonged follow-up time and retrospective analysis may add bias or confounding factors that were not taken into regard.

=>        Thank you for your comment. The retrospective analysis associated with long-term follow-up could lead to several bias but all medical records were analyzed during data compelling to identified potential confounding factors for acute toxicities by regarding post-operative records of patients right before RT treatment beginning, but also for late toxicities especially for heart disease incidental cases. No confounding factors have been found to be associated with incidence of presented acute or late toxicities.

  1. The authors did not provide data on the specific types or severity of acute and late toxicities that were evaluated.

=>        Thank you for your comment. Specific types of acute and late toxicities were recorded and evaluated thanks to CTCAE v.5 for all patients, and graduation of severity of those toxicities were likely graded with the same scale with a range from 0 (no reported toxicity) to 5 (death). “Toxicities were graduated using CTCAE v.5 scale » (line 161).

Round 2

Reviewer 2 Report

Comments and Suggestions for Authors

The author included all the suggested changes and revised the manuscript. 

Comments on the Quality of English Language

Minor editing of English language required

Reviewer 3 Report

Comments and Suggestions for Authors

Accept in present form

Comments on the Quality of English Language

Moderate editing of English language required